# Changes in Health-Related Behaviours and Mental Health in a UK Public Sample during the First Set of COVID-19 Public Health Restrictions

**DOI:** 10.3390/ijerph19073959

**Published:** 2022-03-26

**Authors:** Jason J. Wilson, Lee Smith, Anita Yakkundi, Louis Jacob, Suzanne Martin, Igor Grabovac, Daragh T. McDermott, Rubén López-Bueno, Yvonne Barnett, Laurie T. Butler, Felipe B. Schuch, Nicola C. Armstrong, Mark A. Tully

**Affiliations:** 1Sport and Exercise Sciences Research Institute, School of Sport, Ulster University, Newtownabbey BT37 0QB, UK; jj.wilson@ulster.ac.uk; 2Institute of Mental Health Sciences, School of Health Sciences, Ulster University, Newtownabbey BT37 0QB, UK; 3The Cambridge Centre for Sport and Exercise Sciences, Anglia Ruskin University, Cambridge CB1 1PT, UK; lee.smith@aru.ac.uk; 4Northern Ireland Public Health Research Network, School of Health Sciences, Ulster University, Newtownabbey BT37 0QB, UK; a.yakkundi@ulster.ac.uk; 5Faculty of Medicine, University of Versailles Saint-Quentin-en-Yvelines, 78180 Montigny-le-Bretonneux, France; louis.jacob.contacts@gmail.com; 6Parc Sanitari Sant Joan de Deu/CIBERSAM, Universitat de Barcelona, Fundacio Sant Joan de Deu, Sant Boi de Llobregat, 08830 Barcelona, Spain; 7Centre for Health and Rehabilitation Technologies, Institute of Nursing and Health Research, School of Health Sciences, Ulster University, Newtownabbey BT37 0QB, UK; s.martin@ulster.ac.uk; 8Centre for Public Health, Department of Social and Preventive Medicine, Medical University of Vienna, 1090 Wien, Austria; igor.grabovac@meduniwien.ac.at; 9NTU Psychology, School of Social Sciences, Nottingham Trent University, Nottingham NG1 4FQ, UK; daragh.mcdermott@ntu.ac.uk; 10Department of Physical Medicine and Nursing, University of Zaragoza, 50009 Zaragoza, Spain; rlopezbu@unizar.es; 11School of Life Sciences, Anglia Ruskin University, Cambridge CB1 1PT, UK; yvonne.barnett@aru.ac.uk; 12Faculty of Science & Engineering, Anglia Ruskin University, Cambridge CB1 1PT, UK; laurie.butler@aru.ac.uk; 13Department of Sports Methods and Techniques, Federal University of Santa Maria, Santa Maria 97105-900, Brazil; felipe.schuch@ufsm.br; 14Health and Social Care Research & Development Division, Public Health Agency (Northern Ireland), Belfast BT2 8BS, UK; nicola.armstrong@hscni.net; 15School of Medicine, Ulster University, Londonderry BT48 7JL, UK

**Keywords:** COVID-19 pandemic, health behaviour, social distancing, longitudinal study

## Abstract

Public health restrictions, in response to the COVID-19 pandemic, have had potentially wide-ranging, unintended effects on health-related behaviours such as diet and physical activity and also affected mental health due to reduced social interactions. This study explored how health-related behaviours and mental health were impacted in a sample of the UK public during the first set of COVID-19 public health restrictions. Two online surveys were administered in the UK, one within the first three months of the restrictions (Timepoints 1 (T1—involving pre-pandemic recall) and 2/T2) and another ten weeks later (Timepoint 3/T3). Moderate–vigorous physical activity (MVPA), outdoor time, sitting time, screen time and sexual activity were self-reported. Diet was assessed using the Dietary Instrument for Nutrition Education questionnaire. Mental health was measured using the short-form Warwick–Edinburgh Mental Wellbeing Scale and Becks’ Anxiety and Depression Inventories. Differences between timepoints were explored using the Friedman, Wilcoxon signed-rank, McNemar and McNemar–Bowker tests. Two hundred and ninety-six adults (74% under 65 years old; 65% female) provided data across all timepoints. Between T1 and T2, MVPA, time outdoors and sexual activity decreased while sitting, and screen time increased (*p* < 0.05). Between T2 and T3, saturated fat intake, MVPA, time outdoors, and mental wellbeing increased while sitting, screen time and anxiety symptoms decreased (*p* < 0.05). This study found that depending on the level of COVID-19 public health restrictions in place, there appeared to be a varying impact on different health-related behaviours and mental health. As countries emerge from restrictions, it is prudent to direct necessary resources to address these important public health issues.

## 1. Introduction

The COVID-19 pandemic has led to significant upheaval in citizens’ daily lives across the globe. The rapid worldwide spread of COVID-19 resulted in many countries implementing strict public health restrictions in March/April 2020 to control its spread. These measures included stay-at-home orders, a requirement to practice social distancing and, more recently, a requirement for face coverings when people were out in public for essential purposes such as shopping and caring for vulnerable family members or friends [1]. The unintended consequences of the restrictions are that lengthy periods of social distancing are likely to promote feelings of anxiety, depression and isolation [2] as well as lead to reductions in physical activity and increases in sedentary behaviour [3].

Many cross-sectional studies explored various aspects of health-related behaviours and mental health during the pandemic, including diet [4], physical activity and sedentary behaviour [5,6], sexual health [7,8] and the impact on mental health [9,10]. Whilst there have been fewer longitudinal studies, those published suggest particular aspects of mental health, such as psychological distress, had significantly increased during the initial stages of the COVID-19 pandemic compared to pre-pandemic levels [11]. Other studies were more unequivocal in terms of impacts on mental health [12,13]. In terms of physical activity and sedentary behaviour, there seems to be greater consensus in the literature that these behaviours were negatively impacted [3]. It is important to appreciate that a change in one behaviour (e.g., increased screen-time) is likely to compound changes in other behaviours, such as unhealthier eating and being less active [14]. While some changes in health-related behaviours and mental health may be temporary and recover to pre-pandemic levels once COVID-19 public health restrictions begin to ease, there is the potential that some of these changes may be more permanent. This may negatively impact individuals’ long-term health status, meaning it is important to highlight which health-related behaviours and aspects of mental health are being affected through the continuation of COVID-19 public health restrictions.

This study aimed to explore how numerous health-related behaviours and mental health were impacted during the first set of COVID-19 public health restrictions in the UK. We hypothesised that in comparison to pre-pandemic levels, despite individuals’ health-related behaviours and mental health generally being negatively affected at the outset, as individuals grew accustomed to the situation and as certain restrictions began to ease, most health-related behaviours and different aspects of mental health would make partial recovery to pre-pandemic levels.

## 2. Materials and Methods

### 2.1. Design and Participants

This longitudinal study recruited participants via national media outlets (e.g., BBC news online) and social media websites alongside invitations distributed through existing researcher networks. Eligible participants were UK-based adults aged ≥18 years old. Participants provided their written informed consent after reading an information sheet using a data-encrypted website (i.e., JISC survey platform). All data were anonymous and stored on secure university servers.

The initial online survey was launched in the UK on 17 March 2020 and was available until 11 May 2020, while the second online survey was launched on 28 May 2020 and was available until 26 July 2020. In the first online survey, participants were asked to answer questions related to health-related behaviours and mental health before the COVID-19 pandemic (Timepoint 1/T1) and during the introduction of the first set of COVID-19 public health restrictions (Timepoint 2/T2). At the end of the first online survey, participants were given a choice to be contacted about a follow-up survey. This was not a requirement, and it was made clear that this was optional. If a participant opted in to the follow-up survey, they were asked to provide their email address for this purpose alone. We did not ask for any other identifiable data. The link for the second online survey was emailed out to willing participants approximately 10 weeks later (Timepoint 3/T3). Both surveys were only offered in English. Table 1 highlights the key public health restrictions in place during each timepoint. Anglia Ruskin University Research Ethics Committee provided ethical approval for the study on 16 March 2020.

### 2.2. Data Collection

Demographic information collected included: age (18–24, 25–34, 35–44, 45–54, 55–64 or ≥65 years old), gender (male, female or other), country (England, Scotland, Wales or Northern Ireland), marital status (single/never married, married/domestic partnership, widowed, divorced or separated), numbers living in the household (one, two or over two) and annual household income (<GBP 15,000, GBP 15,000–24,999, GBP 25,000–39,999, GBP 40,000–59,999 or ≥GBP 60,000).

Dietary intakes of fibre, saturated fat and unsaturated fat over the previous week were assessed using the validated Dietary Instrument for Nutrition Education (DINE) questionnaire [16] at T1, T2 and T3. DINE measures fibre and fat consumption across 19 food groups. Higher scores indicate higher fibre intake and higher fat intake. Fibre and saturated fat intake were classified as ‘low’ (<30), ‘medium’ (30–40) or ‘high’ (>40), while the unsaturated fat intake was classified as ‘low’ (<6), ‘medium’ (6–9) or ‘high’ (>9).

With respect to physical activity, participants were asked to self-report how much time they spent on an average day in moderate activity and vigorous activity in hours and minutes. Self-reported moderate and vigorous physical activity were individually truncated to 180 min/day based on established physical activity scoring rules [17] and summed to calculate the number of minutes of moderate–vigorous physical activity (MVPA) per day. A categorical variable (Yes/No) was also developed based on meeting the recent World Health Organisation (WHO) guidelines for physical activity levels of ≥150 min/week [18]. Participants were also asked to recall their average daily time spent outdoors, sitting and watching a screen in hours and minutes. Self-reported outdoor time, sitting time and screen time were all truncated to 960 min/day based on previous recommendations [19]. Categorical variables (Yes/No) for sitting time and screen time were also developed based on a previously used threshold of 480 min/day [20]. Participants were asked how many times they had engaged in sexual activity (e.g., sexual intercourse, masturbation, petting, or fondling) per week. Physical activity, sedentary behaviour and sexual activity questionnaires were completed for T1, T2 and T3.

Mental health, mental wellbeing and loneliness were measured using Beck’s Anxiety Inventory (BAI), Beck’s Depression Inventory (BDI), the short-form Warwick–Edinburgh Mental Wellbeing Scale (SWEMWBS) and the three-item University of California Los Angeles (UCLA) Loneliness Scale. The BAI and BDI both contain 21 items with higher BAI and BDI scores indicating worse anxiety and depressive symptoms. Both BAI and BDI were previously shown to be reliable and valid [21,22]. Scores of ≥16 for the BAI suggest moderate-to-severe anxiety symptoms [23], while scores of ≥20 for the BDI suggest moderate-to-severe depressive symptoms [24]. The SWEMWBS contains seven items and has been validated [25]. Higher scores reflect better mental wellbeing, with scores ≤15.8 indicating poor mental wellbeing [26]. The three-item UCLA Loneliness Scale was shown to be useful in large-scale surveys [27]. Higher scores indicate higher levels of loneliness.

Participants were also asked about their current smoking status (yes or no) and whether they currently consumed alcohol (yes or no).

### 2.3. Statistical Analysis

Analyses were completed using SPSS Version 26 (IBM, Armonk, NY, USA) with continuous data presented as median (25th–75th interquartile range) and categorical data as number (percentage) unless otherwise highlighted. In order to compare demographic characteristics for participants providing valid data at T3 compared with those that did not complete the T3 survey, chi-square tests were performed. Normality testing highlighted that all the health-related behaviours and mental health outcome variables were not normally distributed, which required non-parametric statistical analyses. Friedman tests were used to highlight whether there were any differences between T1, T2 and T3 for the relevant health-related behaviours (diet, physical activity, sedentary behaviour and sexual activity) measured on a continuous scale. Where significant differences were identified, post hoc testing was conducted using Wilcoxon signed–rank tests with Bonferroni correction. Wilcoxon signed–rank tests without correction were used to compare T2 and T3 only (i.e., no T1 data available) for the mental health outcome variables measured on a continuous scale. Where significant differences were identified, then post hoc testing was conducted using Wilcoxon signed–rank tests with Bonferroni correction. In order to examine differences between the timepoints for categorical variables with three levels (e.g., DINE fibre categories low, medium or high), separate McNemar–Bowker tests were completed. When comparing across timepoints for categorical variables with two levels (e.g., drank alcohol (yes or no)), McNemar tests were undertaken. For all of the above statistical tests, Bonferroni corrections were applied if the comparisons involved T1 versus T2 versus T3, resulting in a significance level being set at *p* < 0.017. If the comparison was only between T2 versus T3 (i.e., no T1 data available), then the significance level was set at *p* < 0.05.

## 3. Results

From the original 1087 participants who completed the first online survey covering T1 and T2, 318 participants completed the follow-up survey at T3. However, 22 of these participants did not provide sufficient information to link their responses to the first online survey. Therefore, 296 participants who provided data across all timepoints were included in the final analysis (Table 2). Three-quarters of the sample consisted of adults aged <65 years old, while 65.20% were women. Participants who provided data at T1, T2 and T3 (n = 296) were more likely to be older (χ^2^ = 21.362; *p* < 0.001) and had smaller numbers living in their household (χ^2^ = 15.185; *p* < 0.001) compared with those who did not (n = 791). There were no differences in gender (*p* = 0.657), country (*p* = 0.796), marital status (*p* = 0.241) and annual household income (*p* = 0.984). The median date for completion of the initial online survey was 28 March 2020, while the median date for completion of the second online survey was 4 June 2020.

### 3.1. Diet

In Table 3, there was a significant decrease in DINE fibre scores at T3 versus T1 (Z = −2.584, *p* = 0.010) although there were no significant differences (*p* < 0.017 after Bonferroni adjustment) at T2 versus T1 (*p* = 0.332) or T3 versus T2 (*p* = 0.024). There was a significant increase in DINE saturated fat scores at T3 versus T2 (Z = −2.394, *p* = 0.0167). However, there were no significant differences between T2 and T1 (*p* = 0.026) or T3 and T1 (*p* = 0.939). There was no significant difference in DINE unsaturated fat scores across all time periods (*p* = 0.311). In terms of DINE score categories for fibre, saturated and unsaturated fat intake (i.e., ‘low’, ‘medium’ and ‘high’), there were no significant changes across T1, T2 and T3 (Appendix A).

### 3.2. Physical Activity, Sedentary Behaviour and Sexual Activity

Daily time in moderate–vigorous physical activity (Table 3) significantly decreased at T2 versus T1 (Z = −7.712, *p* < 0.001) and T3 versus T1 (Z = −4.684, *p* < 0.001) while there was a significant increase at T3 versus T2 (Z = −3.297, *p* < 0.001). In terms of meeting the MVPA guidelines, significantly more participants switched to not meeting the MVPA guidelines at T2 and T3 compared with T1 (both *p* < 0.001) than vice versa, but there was no significant change from T2 to T3 (*p* = 0.099) (Appendix A). Similarly, daily time spent outdoors significantly decreased at T2 versus T1 (Z = −8.179, *p* < 0.001), while there was a significant increase in daily time spent outdoors at T3 versus T2 (Z = −8.225, *p* < 0.001). However, there was no significant difference between T3 and T1 (*p* = 0.132).

Daily sitting and screen time (Table 3) both significantly increased at T2 versus T1 (Z = −9.943, *p* < 0.001 and Z = −11.203, *p* < 0.001, respectively) and T3 versus T1 (Z = −4.900, *p* < 0.001 and Z = −7.376, *p* < 0.001, respectively). Daily sitting and screen time both significantly decreased at T3 versus T2 (Z = −5.415, *p* < 0.001 and Z = −3.013, *p* = 0.003, respectively). At T2 compared with T1, more participants (*p* < 0.001) switched to exceeding the 480 min/day threshold for both sitting and screen times (76 and 52 participants, respectively) than vice versa (13 and 5 participants, respectively) (Appendix A). This switch to exceeding the 480 min/day threshold was also evident at T3 versus T1 for both sitting (*p* = 0.004) and screen (*p* < 0.001) time thresholds. However, at T3 versus T2, more participants (*p* < 0.001) switched from exceeding the 480 min/day threshold (60 participants) than vice-versa (19 participants) for sitting time, but there was no significant change for the screen time threshold (*p* = 0.328).

Sexual activity per week (Table 3, Appendix A) significantly decreased at T2 versus T1 (Z = −4.989, *p* < 0.001) and T3 versus T1 (Z = −3.726, *p* < 0.001). However, there was no significant difference between T3 and T2 (*p* = 0.117).

### 3.3. Mental Health

Anxiety scores significantly decreased from T2 to T3 (Z = −3.423, *p* < 0.001) with more participants’ categorised anxiety moving from ‘moderate-severe’ to ‘mild’ at T3 than vice-versa (Table 4, Appendix A). Depression scores did not significantly change from T2 to T3 (*p* = 0.183). Mental wellbeing scores significantly increased from T2 to T3 (Z = −2.419, *p* = 0.016), but there were no significant changes between T2 and T3 for the proportion with ‘poor’ versus ‘average-to-high’ mental wellbeing categories (*p* = 1.000). There were also no significant changes in loneliness scores from T2 to T3 (*p* = 0.188).

### 3.4. Alcohol and Smoking Behaviour

Significantly more participants switched from not drinking alcohol to drinking alcohol at T3 versus T2 (*p* < 0.001) than vice-versa (Table 5). However, there were no significant changes in smoking behaviour (*p* = 1.000).

## 4. Discussion

The findings showed that after the introduction of COVID-19 public health restrictions in the UK, negative, statistically significant changes were reported in time spent in MVPA, time spent outdoors, sitting time, screen time and also sexual activity, with no significant changes in fibre, saturated fat and unsaturated fat intake. Subsequently, as the first set of COVID-19 public health restrictions began to ease, negative changes were still being reported in saturated fat intake and alcohol drinking. However, during the same period, there were positive changes in time spent in MVPA and time spent outdoors, sitting time, screen time, anxiety symptoms and mental wellbeing.

As the first set of COVID-19 public health restrictions was being eased, there was a decrease in fibre intake compared with pre-pandemic levels. This finding is in line with the results of a large survey which highlighted that fruit and vegetable consumption (generally high in fibre) had decreased [28]. However, another study has shown that fibre intake had potentially increased during the pandemic [29]. It is worth noting that this study by Bogataj Jontez and colleagues was much smaller (n = 38) than the current study and measured fibre intake differently. Saturated and unsaturated fat intake appeared to remain largely unaffected. Other research has shown that many individuals had decreased their fat intake through reduced consumption of high-fat and high-sugar foods during the initial stages of the pandemic [30]. Reduced access to certain foods due to panic buying and supply chain issues may have been one reason for this, with other possible contributing factors including job loss, inability to visit shops and reduced household income [31]. Interestingly, saturated fat intake increased from the period that the first set of COVID-19 public health restrictions was introduced up to the point these restrictions began to ease, potentially due to more takeaway restaurants being open again. Less healthy food is generally served in these establishments compared with home-cooked meals [32]. A study in Poland also reported that over half of their sample had reported snacking more during the pandemic [33].

Our study showed that both physical activity and sedentary behaviour at the point when COVID-19 public health restrictions began to ease had not recovered to pre-pandemic levels in terms of daily time spent in MVPA, sitting and screen time as well as meeting the recommended physical activity guidelines/suggested sedentary behaviour thresholds. At the start of T3, examples of COVID-19 public health restrictions being relaxed included allowing larger indoor and outdoor gatherings. It is important to note further relaxations to the COVID-19 public health restrictions took place throughout T3 (e.g., more non-essential retailers were allowed to reopen), and there was even a UK-wide initiative put in place to encourage the visiting of cafes and restaurants again (i.e., Eat Out to Help Out). Our findings are in agreement with a recent systematic review of 64 studies, which highlighted that in most cases, the COVID-19 public health restrictions caused reductions in physical activity accompanied by higher levels of sedentary behaviour [3]. It is important to note that certain groups are likely to have been more negatively impacted compared to others. For example, reduced time spent in physical activity was highlighted in populations with medical conditions such as Type II diabetes and osteoarthritis [3]. In addition, young adults and those not married were shown to have had greater reductions in physical activity, coupled with increases in sedentary behaviour, compared with other population groups [6,34]. A US-based cross-sectional study even highlighted those achieving the physical activity guidelines before COVID-19 were most likely to see significant drops in physical activity time during the initial stages of the pandemic compared with those not meeting the physical activity guidelines [5]. With the necessity for many workers to commence and continue working from home, it is wholly unsurprising that screen time has remained high, even as the first set of COVID-19 public health restrictions were eased [34]. Whereas most work meetings would have traditionally taken place face-to-face, many have now moved to virtual settings using platforms such as Zoom and Microsoft Teams.

From the introduction of the first set of COVID-19 public health restrictions until the point in which these restrictions had begun to be eased, anxiety symptoms decreased and mental wellbeing increased while depressive symptoms and levels of loneliness were not significantly altered. These results are supported by the relevant literature from other countries [12,13,35,36]. The improvement in anxiety symptoms and mental wellbeing could be due to increased familiarity with the COVID-19 restrictions as well as potentially adopting simple coping behaviours, such as sticking to a set routine, reducing news consumption surrounding the pandemic and performing more home-based cooking [37]. However, it is important to note that certain subgroups, such as those with pre-existing physical and mental health conditions as well as those from more socially disadvantaged backgrounds, are more likely to be impacted than others [12,13,35].

Sexual activity appeared to decrease during the first set of COVID-19 public health restrictions, and weekly levels during the point these restrictions began to ease still have not returned to pre-pandemic levels. Our previous cross-sectional study on sexual activity during the pandemic compared the initial stages with pre-pandemic [8]. Our current study builds on this by showing that sexual activity remained below pre-pandemic levels. This is supported by other research in Europe [38] and Asia [7]. This is an important finding as healthy sexual activity was shown to be beneficial for psychological and relational health during the pandemic [39]. In terms of smoking behaviour, it was positive to see that there was no increased number of individuals smoking during the pandemic. Another UK-based study found that cigarette smoking actually decreased during the pandemic [11]. This is important considering smoking is suggested to be associated with increased progression of adverse COVID-19 outcomes [40]. The current study also found that more individuals had started drinking alcohol when the first set of COVID-19 public health restrictions had begun to ease compared to the initial stages when full restrictions were in place. Another study also supports this finding, with a possible reason being that some could be potentially using alcohol as a coping mechanism due to a lack of recreational and social activities as the pandemic has continued on [11].

One of the key strengths of this current study is the variety of health-related behaviours and different aspects of mental health, which were concurrently assessed across multiple timepoints in a UK sample during the first set of COVID-19 public health restrictions. Another strength of this study was the use of validated questionnaires for measuring dietary intake and mental health in the general population. However, study limitations must also be considered. While the initial survey was completed by 1087 participants, the follow-up survey was only completed by 318 participants; 296 of these provided useable data. This is likely to result in selection bias and reduce the power of the findings. Those completing the follow-up survey were also different in terms of being older and living in smaller households. With the survey being conducted online, self-report and recall biases are likely to have been introduced. However, this was the most practical method given the COVID-19-related restrictions in place. As participants were required to recall their diet, physical activity, sedentary behaviour and sexual activity from before the pandemic, this is likely to have introduced recall bias. Finally, because data collection for each timepoint took place over several months, it is likely that different levels of public health restrictions were in place throughout each timepoint. However, it is worth noting that most participants in the sample completed the first online survey within one month of the first full set of COVID-19 public health restrictions.

## 5. Conclusions

In summary, this study found that depending on the level of COVID-19 public health restrictions in place, there appeared to be a varying impact on different health-related behaviours and mental health. These findings have important public health implications as they highlight the health-related behaviours and aspects of mental health, which may have improved since the first public health restrictions were introduced while highlighting others still being negatively affected. In terms of some practical implications of these findings for society, it is important for individuals to consider strategies focused on decreasing their sedentary behaviour as well as increasing their time spent taking part in moderate–vigorous physical activity. In addition, it is important for individuals to consider increasing their fibre intake as well as moderating their alcohol intake. With key aspects of life such as home-based working, travel, interactions with different people and shopping habits likely to be changed on a permanent basis, along with the possibility of further restrictions to control the spread of new variants of the virus, it is important to conduct future research which continues to monitor the situation in terms of these important health-related behaviours and aspects of mental health in order to direct public health policy appropriately.

## Figures and Tables

**Table 1 ijerph-19-03959-t001:** COVID-19 public health restrictions applied during different study timepoints (adapted from [15]).

Month (Timepoint)	Summary of Public Health Restrictions
First half of March (T1)	No public health restrictions being implemented
Second half of March (T2)	First full set of COVID-19 public health restrictions was introduced: People advised to stay at home (only permitted to leave for essential reasons only); indoor and outdoor social gatherings banned; non-essential high street business closures; social distancing of 2 m; school closures.
April (T2)	First full set of COVID-19 public health restrictions was still being implemented.
First half of May (T2)	First full set of COVID-19 public health restrictions was still being implemented.
Second half of May (T3)	Those who could not work from home were advised to return to their workplace but not use public transport to do so. Outdoor recreation was allowed in groups of up to six people. Other COVID-19 public health restrictions remain.
June (T3)	Some COVID-19 public health restrictions were relaxed: Stay-at-home message was replaced with a requirement to be home overnight; limited outdoor social gatherings allowed; some non-essential high street businesses allowed to reopen; phased reopening of schools and relaxing of 2 m social distancing rule (in England only).
July (T3)	More COVID-19 public health restrictions were relaxed: Larger outdoor social gatherings were allowed; limited indoor gatherings were allowed; other non-essential high street businesses were allowed to reopen (e.g., hairdressers, gyms and spa facilities).

Abbreviations: T1 = timepoint 1; T2 = timepoint 2; T3 = timepoint 3.

**Table 2 ijerph-19-03959-t002:** Sample demographic characteristics of the 296 participants providing data at T1, T2 and T3.

Characteristics	Number (%)
Age	
18–24 years old	27 (9.12)
25–34 years old	44 (14.86)
35–44 years old	43 (14.53)
45–54 years old	58 (19.59)
55–64 years old	47 (15.88)
≥65 years old	76 (25.68)
Not reported	1 (0.34)
Gender	
Male	98 (33.11)
Female	193 (65.20)
Other	4 (1.35)
Not reported	1 (0.34)
Country	
England	234 (79.05)
Scotland	7 (2.36)
Wales	3 (1.01)
Northern Ireland	50 (16.89)
Not reported	2 (0.68)
Marital status	
Single or never married	89 (30.07)
Married or domestic partnership	164 (55.41)
Widowed	13 (4.39)
Divorced	24 (8.11)
Separated	4 (1.35)
Not reported	2 (0.68)
Numbers living in household	
One	66 (22.30)
Two	131 (44.26)
Three or more	98 (33.11)
Not reported	1 (0.34)
Annual household income	
<GBP 15,000	45 (15.20)
GBP 15,000–24,999	55 (18.58)
GBP 25,000–39,999	69 (23.31)
GBP 40,000–59,999	60 (20.27)
≥GBP 60,000	64 (21.62)
Not reported	3 (1.01)

Abbreviations: SD = standard deviation.

**Table 3 ijerph-19-03959-t003:** Changes in diet, physical activity, sedentary behaviour and sexual activity during the first set of COVID-19 public health restrictions.

Variables	T1 Median (25th–75th IQR)	T2 Median (25th–75th IQR)	T3 Median (25th–75th IQR)	Friedman Test Difference
DINE				
Fibre intake score, n = 296	32.0 (26.0–39.0) ^b^	32.0 (25.0–40.0)	30.0 (24.0–38.0)	*p* = 0.032 *
Saturated fat intake score, n = 296	22.0 (18.0–26.0)	21.0 (17.0–26.0) ^c^	22.0 (18.0–27.0)	*p* = 0.012 *
Unsaturated fat score, n = 296	9.0 (7.0–11.0)	9.0 (7.0–11.0)	9.0 (6.0–11.0)	*p* = 0.311
Physical activity and sedentary behaviour				
MVPA time (min/day), n = 287 ^d^	120.0 (60.0–180.0) ^a,b^	60.0 (30.0–135.0) ^c^	90.0 (35.0–150.0)	*p* < 0.001 *
Outdoor time (min/day), n = 285 ^e^	120.0 (90.0–240.0) ^a^	60.0 (30.0–135.0) ^c^	120.0 (60.0–240.0)	*p* < 0.001 *
Sitting time (min/day), n = 276 ^f^	360.0 (273.8–540.0) ^a,b^	517.5 (360.0–720.0) ^c^	480.0 (300.0–600.0)	*p* < 0.001 *
Screen time (min/day), n = 293 ^g^	240.0 (120.0–360.0) ^a,b^	360.0 (240.0–540.0) ^c^	300.0 (180.0–525.0)	*p* < 0.001 *
Sexual activity				
Weekly sexual activity, n = 272 ^h^	1.0 (0.0–2.0) ^a,b^	0.0 (0.0–1.0)	0.0 (0.0–2.0)	*p* < 0.001 *

Abbreviations: DINE = Dietary Instrument for Nutrition Education; IQR = interquartile range; MVPA = moderate–vigorous physical activity time; T1 = timepoint 1; T2 = timepoint 2; T3 = timepoint 3. * = Significant difference (*p* < 0.05). ^a^ = Significant difference (*p* < 0.017 after Bonferroni adjustment) T1 vs T2. ^b^ = Significant difference (*p* < 0.017 after Bonferroni adjustment) T1 vs. T3. ^c^ = Significant difference (*p* < 0.017 after Bonferroni adjustment) T2 vs. T3. ^d^ = 9 participants (3.04% of the total sample) did not report these data. ^e^ = 11 participants (3.72% of the total sample) did not report these data. ^f^ = 20 participants (6.76% of the total sample) did not report these data. ^g^ = 3 participants (1.01% of the total sample) did not report these data. ^h^ = 24 participants (8.11% of the total sample) did not report these data.

**Table 4 ijerph-19-03959-t004:** Changes in mental health during the first set of COVID-19 public health restrictions.

Variables	T2 Median (25th–75th IQR)	T3 Median (25th–75th IQR)	Wilcoxon Signed-Rank Test Difference
BAI score, *n* = 296	7.0 (3.0–19.0)	6.0 (2.0–15.0)	*p* < 0.001 *
BDI score, *n* = 296	8.0 (4.0–16.0)	7.5 (3.0–14.0)	*p* = 0.183
SWEMWBS score, *n* = 296	20.7 (18.0–24.1)	21.5 (18.1–24.1)	*p* = 0.016 *
UCLA loneliness score, *n* = 293 ^a^	5.0 (3.0–6.0)	5.0 (3.0–6.0)	*p* = 0.188

Abbreviations: BAI = Beck’s Anxiety Inventory; BDI = Beck’s Depression Inventory; DINE = Dietary Instrument for Nutrition Education; IQR = interquartile range; MVPA = moderate–vigorous physical activity time; SWEMWBS-7 = Short Warwick–Edinburgh Mental Wellbeing Scale; T2 = timepoint 2; T3 = timepoint 3. * = Significant difference (*p* < 0.05). ^a^ = 3 participants (1.01% of the total sample) did not report these data.

**Table 5 ijerph-19-03959-t005:** Changes in alcohol and smoking behaviours during the first set of COVID-19 public health restrictions.

	**Drank Alcohol at T3**		**Count, *n***	***p*-Value**
**Drank alcohol at T2**	Yes	No		
Yes	197	2	293 ^a^	*p* < 0.001 *
No	19	75		
	**Smoking at T3**		**Count, *n***	***p*-Value**
**Smoking at T2**	Yes	No		
Yes	20	4	293 ^a^	*p* = 1.000
No	4	265		

Abbreviations: T2 = timepoint 2; T3 = timepoint 3. * = Significant difference (*p* < 0.05). ^a^ = 3 participants (1.01% of the total sample) did not report these data.

## Data Availability

The data presented in this study are available on request from the corresponding author.

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
