# Peer review of "Changes in Health-Related Behaviours and Mental Health in a UK Public Sample during the First Set of COVID-19 Public Health Restrictions"

_ijerph, 2022, doi:10.3390/ijerph19073959_

Round 1

Reviewer 1 Report

Dear author,   First of all, congratulations on the work you have done and the effort that it takes to effort it takes to conduct research of this nature.   It provides good design, good analysis and discussion along with limitations.   Congratulations on the work done.   However, I would suggest a number of recommendations:  

-Expand the introduction.  

-Incorporate what theoretical implications this work has for the scientific community.   

-Incorporate what practical implications this work has for the community/society/population.

 -Incorporate the strengths of your work.

Yours sincerely.

Author Response

Dear Reviewer 1,

Thank you very much for your helpful comments, these have certainly helped us to strengthen our paper. Please see below for responses to your queries.

Comment 1: Dear author, First of all, congratulations on the work you have done and the effort that it takes to effort it takes to conduct research of this nature.   It provides good design, good analysis and discussion along with limitations.   Congratulations on the work done.   However, I would suggest a number of recommendations: 

Response 1: Thank you for this comment, hopefully we have addressed your recommendations.

Comment 2: Expand the introduction. 

Response 2: We have added extra detail to the “Introduction” section (lines 75-81). The reviewer has not identified what aspect of the Introduction required expansion, so we trust this meets your expectation.

Comment 3: Incorporate what theoretical implications this work has for the scientific community.  

Response 3: There are a range of approaches to research, derived from a variety of epistemological perspectives. Whilst we fully appreciate and often utilise theoretical frameworks to derive models for testing, this is not the approach taken in this project. This study was developed in the very early stages of the first wave of the pandemic, when little was known as to the course it would take. Therefore, we employed observational epidemiological methods to capture potential impacts of the lockdown on health related behaviours. We feel the work in this paper has less theoretical implications but is more focused on applied outcomes to record any changes in health-related behaviours and aspects of mental health which have been more impacted due to the public health restrictions, in order to inform future public health approaches to dealing with potential changes longer term.

Comment 4: Incorporate what practical implications this work has for the community/society/population.

Response 4: We have added the following sentences to the “Conclusions” section: “In terms of some practical implications of these findings for society, it will be important for individuals to consider strategies focused on decreasing their sedentary behaviour as well as increasing their time spent taking part in moderate-vigorous physical activity. In addition, it will be important for individuals to consider increasing their fibre intake as well as moderate their alcohol intake.”

Comment 5: Incorporate the strengths of your work.

Response 5: We have now added an additional strength in that we used validated questionnaires for measuring dietary intake and mental health.

Reviewer 2 Report

The authors did not use IJERPH template and the whole manuscript document needs to be corrected according to the instructions for the authors.

The authors mentioned the T1 time before the COVID-19 pandemic, but they stated that the survey was launched on 17th March 2020. Which statement is accurate, the authors should elaborate more.

The authors should state the ethical approval ID.

How the data collection was done? How anonymity was preserved when participants needed to repeatedly enter the survey?

There are a very large number of univariate analyses in this manuscript, but no attempt to conduct multivariable analyses to account for the effect of the
plethora of variables and potential multicollinearity issues. I suggest
using the univariate analyses as a basis for building multivariable
logistic regression models using either forward selection or backward
elimination procedures.

What data was mandatory to fill in because there are different n for each characteristic? Did the authors exclude incomplete surveys?

Table 2. - The sum of percentages needs to be 100%.

There is no conclusion section in this manuscript and it is mandatory to have one.

References need correction according to the instructions for the authors.

Author Response

Dear Reviewer 2,

Thank you very much for your helpful comments, these have certainly helped us to strengthen our paper. Please see below for responses to your queries.

Comment 1: The authors did not use IJERPH template and the whole manuscript document needs to be corrected according to the instructions for the authors.

Response 1:  According to the instructions for authors, the use of the IJERPH template is optional. However, in response to this request, the template has now been completed and the manuscript follows the journal’s specifications.

Comment 2: The authors mentioned the T1 time before the COVID-19 pandemic, but they stated that the survey was launched on 17th March 2020. Which statement is accurate, the authors should elaborate more.

Response 2: We have clarified in the manuscript that participants were asked to answer questions related to health-related behaviours and mental health before the COVID-19 pandemic (Timepoint 1/T1) and during the introduction of the first set of COVID-19 public health restrictions (Timepoint 2/T2) in the first online survey.

Comment 3: The authors should state the ethical approval ID.

Response 3: Anglia Ruskin University Research Ethics Committee do not provide ID numbers for approved ethics applications. We have given the date of the approval to be as specific as we can.

Comment 4: How the data collection was done? How anonymity was preserved when participants needed to repeatedly enter the survey?

Response 4: The survey data was collected online using the JISC survey platform. All data was anonymous, and stored on secure university servers. At the end of the first survey, participants were given the choice to be contacted about a follow-up survey. This was not a requirement, and it was made clear that this was optional. If a participant opted in to follow-up surveys, they were asked to provided their email address for this purpose alone. We did not ask for any other identifiable data. These sentences have been incorporated within the first and second paragraphs in the Methods section.

Comment 5: There are a very large number of univariate analyses in this manuscript, but no attempt to conduct multivariable analyses to account for the effect of the plethora of variables and potential multicollinearity issues. I suggest using the univariate analyses as a basis for building multivariable logistic regression models using either forward selection or backward elimination procedures.

Response 5: This is a valid point and one we considered in developing the analysis plan, However, for the current analyses, we decided not to treat the behaviours as potentially inter-dependant. There is insufficient evidence in the literature to support this. We strongly feel there would not be a sound rationale for combining some behaviours. Therefore, we did not combine them in a multivariable model to examine their independent effects, and as such do not present the results as independent of one another. However, we did apply Bonferonni corrections to account for multiple hypothesis testing.

Comment 6: What data was mandatory to fill in because there are different n for each characteristic? Did the authors exclude incomplete surveys?

Response 6: Thank you for this query. We can clarify only the consent form data was mandatory to fill in whereas the rest of the questionnaire was optional, which is the reason for small amounts of missing data for some characteristics / variables. Where there were some sections not completed, we did not exclude the whole record in an effort to maximise our sample size.

Comment 7: Table 2 - The sum of percentages needs to be 100%.

Response 7: Marital status adds up to 99.99% due to 2 d.p. being used. We feel going lower than 2 d.p. would not provide any more usefulness.

Comment 8: There is no conclusion section in this manuscript and it is mandatory to have one.

Response 8: Thank you for recommending this, we have now added a “Conclusions” section.

Comment 9: References need correction according to the instructions for the authors.

Response 9: Again, thank you for this comment. These have now been updated to the specifications of the journal.

Reviewer 3 Report

The manuscript covers a topic that is of high public health relevance and interest and is generally well written. I have a few comments/queries listed here:

Material and Methods

  • In line 91, the authors state that the second online survey was launched on May 28th 2020. Till when was it available? Where reminders sent out?
  • Consider replacing 'if' in line 151 with 'whether' and in lines 153 and 156 with 'where'.
  • line 162 - should it not be "Bonferroni correction were applied"?
  • A general question: Was the study advertised on the different channels in different languages and was the survey multi-lingual?

Results

  • Consider using sub-headings for each sub-section to help structure your findings and guide the reader.
  • Perhaps start by saying how many people participated at T1 and T2, especially as you go on to give the loss to follow-up as a limitation in the discussion.
  • The beginning of line 180 is somewhat misleading as some of the findings stated in lines 180-184 are not to be found in Table 3 - e.g., the Z and p values for the change in DINE fibre scores at T3 vs T1, and the Z and p values for the increase in DINE saturated fat scores at T3 vs T2.
  • Consider restructuring sentence in lines 171-173 such that it basically says "Participants who provided T1, T2 and T3 data were more likely to be older than those who did not, and also had smaller households."
  • In line 199, consider adding 'participants' between more and switched
  • The beginning of the sentence starting with "This was a similar pattern at T3 vs T1", lines 211-212, needs rewording.
  • Consider restructuring Table 2 so that it has 2 columns with each category adding up to 296 as stated in the table heading.  The information presented in the column 'category' can be presented under 'characteristics'. Where applicable, missing should be stated. The table heading also needs to be more information, e.g., Sample demographic characterstics of the 296 participants who provided data at T1, T2 and T3 of the survey.
  • The authors collected data on various demographic characteristics, none of which appear to have been considered in the analyses. Is there any particular explanation for this? Further, data on nationality/ethnicity/migration are not provided. Was this information not collected?

Discusson

  • In lines 253-255, the authors state that reduced access to ceratin may have have been one reason for the decrease in fat intake, and go on to give other possible reasons, e.g., inability to visit shops. I'm wondering if reduced access to the foods and inability to visit shops are not basically one and the same thing?
  • In lines 272-276, the authors state that certain groups are more likely to have been negatively affected than others and go on to give the example of those achieving physical activity guidelines before COVID-19. While this might be true, I would have expected them to cite examples that point out inequalities/inequities based on sociodemographic characteristics.
  • The authors should provide a bit more detail to explain what they mean in lines 290-291 - certain subgroupy being more likely to be impacted than others.
  • The sentence beginning "Fibre intake decreased.." in lines 245-246 needs rewording to make the meaning clearer. The authors should also consider rewording the beginning of the sentence after that, perhaps say "This finding is in line with results of a large survey...".
  • In 249, it might be helpful to add the number of participants of the "much smaller" study the authors refer to. In the following sentence (line 250), it is not clear which study the authors are referring to: their own, or the 'much smaller' one?
  • In line 320, consider replacing 'recollect' with 'recall'.

Author Response

Dear Reviewer 3,

Thank you very much for your helpful comments, these have certainly helped us to strengthen our paper. Please see below for responses to your queries.

Comment 1: The manuscript covers a topic that is of high public health relevance and interest and is generally well written. I have a few comments/queries listed here:

Response 1: Thank you for this nice comment. We hope to have sufficiently addressed your queries.

Material and Methods

Comment 2: In line 91, the authors state that the second online survey was launched on May 28th 2020. Till when was it available? Where reminders sent out?

Response 2: We have clarified it was available until 26th July 2020. We have also clarified that willing participants who completed the first online survey were emailed a link to complete the second online survey from 28th May 2020.

Comment 3: Consider replacing 'if' in line 151 with 'whether' and in lines 153 and 156 with 'where'.

Response 3: These words have been replaced as recommended.

Comment 4: line 162 - should it not be "Bonferroni correction were applied"?

Response 4: This phrasing has now been updated as recommended.

Comment 5: A general question: Was the study advertised on the different channels in different languages and was the survey multi-lingual?

Response 5: The survey was only advertised and offered in the English language. This has been clarified near the end of the “Design and participants” section.

Results

Comment 6: Consider using sub-headings for each sub-section to help structure your findings and guide the reader.

Response 6: As recommended, sub-headings have been used in the Results section to aid the reader.

Comment 7: Perhaps start by saying how many people participated at T1 and T2, especially as you go on to give the loss to follow-up as a limitation in the discussion.

Response 7: The first sentence in the “Results” section has now been updated to clarify how many took part in the original online survey covering T1 and T2 (i.e. n=1087).

Comment 8: The beginning of line 180 is somewhat misleading as some of the findings stated in lines 180-184 are not to be found in Table 3 - e.g., the Z and p values for the change in DINE fibre scores at T3 vs T1, and the Z and p values for the increase in DINE saturated fat scores at T3 vs T2.

Response 8: Table 3 shows the direction of the change in fibre and saturated / unsaturated fat intake between the different time points and highlights where the significant differences lie (e.g. a,b,c). However, we felt adding in the Z scores and p values for each comparison would have made this particular table too cluttered and harder to interpret, which is why this information has been added to the text instead. The text clarifies the direction of change shown in this table.

Comment 9: Consider restructuring sentence in lines 171-173 such that it basically says "Participants who provided T1, T2 and T3 data were more likely to be older than those who did not, and also had smaller households."

Response 9: The sentence has now been restructured as recommended.

Comment 10: In line 199, consider adding 'participants' between more and switched

Response 10: “Participants” has been added as recommended.

Comment 11: The beginning of the sentence starting with "This was a similar pattern at T3 vs T1", lines 211-212, needs rewording.

Response 11: This has now been reworded to: “This switch to exceeding the 480 minutes/day threshold was also evident at…”.

Comment 12: Consider restructuring Table 2 so that it has 2 columns with each category adding up to 296 as stated in the table heading.  The information presented in the column 'category' can be presented under 'characteristics'. Where applicable, missing should be stated. The table heading also needs to be more information, e.g., Sample demographic characterstics of the 296 participants who provided data at T1, T2 and T3 of the survey.

Response 12: As recommended, the table has been reduced to 2 columns, missing data has been clarified as a footnote and the title has now been updated to include more information.

Comment 13: The authors collected data on various demographic characteristics, none of which appear to have been considered in the analyses. Is there any particular explanation for this? Further, data on nationality/ethnicity/migration are not provided. Was this information not collected?

Response 13: Regarding the first point, we feel this is an interesting consideration. However, the type of statistical analyses we conducted would not lend itself to adjusting for the demographic characteristics collected. Also, stratifying our analyses by demographic categories would mean very high numbers of statistical models needing to be undertaken and would result in fragmented sample sizes, resulting in comparisons between much smaller groups. With the current paper already containing a lot of analyses, we feel adding more would confuse the message from this particular paper. Regarding the second point, we have highlighted the participants’ residing country in Table 2 but we did not collect data on ethnicity / migration status in this survey.

Discussion

Comment 14: In lines 253-255, the authors state that reduced access to ceratin may have have been one reason for the decrease in fat intake, and go on to give other possible reasons, e.g., inability to visit shops. I'm wondering if reduced access to the foods and inability to visit shops are not basically one and the same thing?

Response 14: We can clarify that the first reason regarding reduced access was referring to certain foods not being as widely available, due to panic buying and supply chain issues whereas the second reason refers more to individuals being afraid / being unable to physically visit shops. This has now been clarified within this sentence.

Comment 15: In lines 272-276, the authors state that certain groups are more likely to have been negatively affected than others and go on to give the example of those achieving physical activity guidelines before COVID-19. While this might be true, I would have expected them to cite examples that point out inequalities/inequities based on sociodemographic characteristics.

Response 15: Thank you for this helpful comment. In response, we have added two additional sentences highlighting how other groups such as those with medical conditions, younger adults and those not being married have had significant reductions in physical activity compared to other populations.

Comment 16: The authors should provide a bit more detail to explain what they mean in lines 290-291 - certain subgroup being more likely to be impacted than others.

Response 16: We have now elaborated further by highlighting subgroups which have been more likely to have been impacted such as those with pre-existing physical and mental health conditions as well as those from more socially disadvantaged backgrounds.

Comment 17: The sentence beginning "Fibre intake decreased.." in lines 245-246 needs rewording to make the meaning clearer. The authors should also consider rewording the beginning of the sentence after that, perhaps say "This finding is in line with results of a large survey...".

Response 17: This sentence has now been reworded to make It clearer to read and we have also added in the extra wording at the beginning of the next sentence as recommended.

Comment 18: In 249, it might be helpful to add the number of participants of the "much smaller" study the authors refer to. In the following sentence (line 250), it is not clear which study the authors are referring to: their own, or the 'much smaller' one?

Response 18: This sentence has now been written more clearly, highlighting the referenced study by Bogataj Jontez and colleagues had 38 participants.

Comment 19: In line 320, consider replacing 'recollect' with 'recall'.

Response 19: The word has been updated as recommended.

Round 2

Reviewer 2 Report

The authors substantially improved their manuscript. They also very well answered my all questions.

But, there are a few things to do. Still, they didn't use IJERPH word template available on the webpage of the manuscript.

Also, the authors should clearly report missing values, in all tables, to know how many valid values are used in the analysis and how many (percentage) were excluded because of missing values.

Author Response

Dear Reviewer 2,

Please see below for responses to your further queries.

Comment 1: The authors substantially improved their manuscript. They also very well answered my all questions.

Response 1: Again, thank you very much for your helpful comments and glad we addressed these sufficiently. These suggestions have really helped to improve our paper.

Comment 2: But, there are a few things to do. Still, they didn't use IJERPH word template available on the webpage of the manuscript.

Response 2: The Assistant Editor has told us: “Please note that you do not have to use the IJERPH template as the reviewer suggested as we will help to apply the template before the paper is published.”

Comment 3: Also, the authors should clearly report missing values, in all tables, to know how many valid values are used in the analysis and how many (percentage) were excluded because of missing values.

Response 3: This has now been done in Tables 2-5.